# Serendipity in Neuro-Oncology: The Evolution of Chemotherapeutic Agents

**DOI:** 10.3390/ijms26072955

**Published:** 2025-03-25

**Authors:** Denise Nadora, Shawyon Ezzati, Brandon Bol, Orwa Aboud

**Affiliations:** 1College of Medicine, California Northstate University, Elk Grove, CA 95757, USA; shawyon.ezzati9555@cnsu.edu (S.E.); brandon.bol10919@cnsu.edu (B.B.); 2Department of Neurology, Comprehensive Cancer Center, University of California, Davis, CA 95616, USA; oaboud@ucdavis.edu; 3Department of Neurological Surgery, Comprehensive Cancer Center, University of California, Davis, CA 95616, USA

**Keywords:** serendipitous drug discovery, Chemotherapy in CNS tumors, drug repurposing in oncology

## Abstract

The development of novel therapeutics in neuro-oncology faces significant challenges, often marked by high costs and low success rates. Despite advances in molecular biology and genomics, targeted therapies have had limited impact on improving patient outcomes in brain tumors, particularly gliomas, due to the complex, multigenic nature of these malignancies. While significant efforts have been made to design drugs that target specific signaling pathways and genetic mutations, the clinical success of these rational approaches remains sparse. This review critically examines the landscape of neuro-oncology drug discovery, highlighting instances where serendipity has led to significant breakthroughs, such as the unexpected efficacy of repurposed drugs and off-target effects that proved beneficial. By exploring historical and contemporary cases, we underscore the role of chance in the discovery of impactful therapies, arguing that embracing serendipity alongside rational drug design may enhance future success in neuro-oncology drug development.

## 1. Introduction

Neuro-oncology is a complex field focused on the diagnosis and treatment of brain tumors and neoplasms originating from the nervous system. Despite years of extensive research, neurological cancers remain difficult to treat due to the tumors’ locations; the body’s natural defense system, which inhibits the delivery of medications; and the inherent complexities of these tumors [1]. One of the most significant obstacles when treating these neoplasms is the blood–brain barrier (BBB), preventing many chemotherapy drugs from reaching the tumor [2]. Generally, compounds that are able to cross the BBB are small, lipophilic molecules that are able to passively diffuse across the membrane [3]. However, the tumors themselves are heterogenous and therefore are affected by these drugs differently, making the treatment more challenging [4]. As a result, neuro-oncology heavily relies on a multidisciplinary approach to the treatment of malignancies, including immunotherapy, radiation, surgery, and chemotherapy [5].

Prior to the use of medications as a therapeutic option, conventional treatments focused on surgical resection and radiation. The discovery of chemotherapy came from the unfortunate events of World War II, when a German cargo carrying mustard gas canisters exploded, and autopsies of exposed military soldiers showed damage in rapidly dividing white blood cells [6]. This led researchers to speculate that this concept could be applied to cancer cells, which are also able to proliferate rapidly [6]. This accidental discovery opened new avenues for cancer treatment. In the complex world of medical research, serendipity has played an important role in some of the most groundbreaking discoveries. Many medications have been used for centuries prior to their discovery as chemotherapy and were even originally developed for other uses. Since then, they have proven to be unexpectedly effective in targeting and treating malignancies. These serendipitous discoveries not only highlight the unpredictable nature of medical research but also lay the groundwork for future innovations in the treatment of neurological cancers. Here, we discuss clinically noteworthy, serendipitous discoveries in neuro-oncology. The timeline of FDA approvals for the conventional chemotherapeutic agents is summarized in Figure 1.

## 2. Serendipitous Discovery of Conventional Therapies

### 2.1. Methotrexate 

Methotrexate is a folate inhibitor widely used in autoimmune diseases and cancers [7]. Folate is necessary for cell growth. It supports deoxyribonucleic acid (DNA) synthesis, repair, and methylation, which are essential for proliferation and survival of rapidly dividing cells like cancer cells [8,9,10]. Methotrexate was first discovered by Dr. Sidney Farber, who believed that inhibiting folate would slow the rapid division of cancer cells. When methotrexate was administered to patients with acute lymphoblastic leukemia (ALL) in 1948, it elicited an improvement of symptoms and temporary remission [11]. This sparked more studies into its therapeutic effects in cancer. Dr. Roy Hertz and Dr. Min Li showed methotrexate’s efficacy in treating gestational choriocarcinoma. Dr. Hertz was studying the effects of folate inhibition on dividing female reproductive cells when he heard of Dr. Li using methotrexate to treat metastatic melanoma. Dr. Li was not successful in treating melanoma but found significant decreases in beta-human chorionic gonadotropin (B-HCG) in his patients. This was promising to Dr. Hertz, who tried and found success in treating gestational choriocarcinoma with methotrexate. This was the first time a solid tumor responded to chemotherapy [12,13,14,15].

After methotrexate was unexpectedly found to be effective in cancer treatment, its application was extended to central nervous system (CNS) malignancies, uncovering a new and valuable therapeutic use. Until this point, treatments of primary CNS lymphoma had been ineffective because of failure of drug to cross the blood–brain barrier [16,17,18]. Methotrexate was the first chemotherapy drug to penetrate it and offer improvement in survival [19,20,21,22]. Treatment for primary CNS lymphoma remains imperfect, as there is no regimen that offers consistent resolution or therapeutic benefit across all populations. However, methotrexate remains a standard drug in the different regimens tried and has offered response rates of up to 70%, meaning that up to 70% of patients experience a reduction in tumor size or complete elimination of cancer. Additionally, it has led to improvements in 2-year progression-free survival rates, with up to 50% of patients remaining free from disease progression over that period. Since the malignancy is complex in nature, treatment continues to be refined.

Most recently, methotrexate was tried with rituximab and cytarabine, which was found to have an even greater therapeutic benefit than methotrexate alone [23,24,25,26,27]. The current treatment of primary CNS lymphoma includes combination chemotherapy with methotrexate, surgery, and radiation, offering successful long-term disease control in half of newly diagnosed patients [27,28]. The discovery of methotrexate’s success in the treatment of primary CNS lymphoma launched the development of regimens that have allowed for control of a malignancy that was previously uncontrollable. 

### 2.2. Vinblastine 

Vinblastine, known commercially as Velban, is naturally derived from a plant from Madagascar known as the pink periwinkle (*Catharanthus roseus*) [29,30]. Physician–scientist Robert Noble and organic chemist Charles Beer discovered vinblastine, the first of the “vinca alkaloid” family of chemotherapy [29]. While today we commonly acknowledge vinblastine to be a staple drug in our arsenal of chemotherapeutic agents, it was not originally utilized for its anticancer properties; vinblastine was first thought to be useful in treating diabetes [30]. It has been historically noted that Dr. Robert Noble’s brother, Clark Noble, was seeing a patient that had recently visited Jamaica and brought him back *C. roseus* leaves, and she described how the natives in Jamaica had used this plant to treat diabetes for generations [30]. Dr. Noble’s intellectual curiosity took hold after learning from his brother about the potential medical use of the pink periwinkle plant, and he began to test vinblastine’s effect on the blood glucose levels in rats [30]. However, its initial tests yielded low effects on blood glucose levels.

Vinblastine was then tested to see if it had any antibiotic potential in rats infected with *Pseudomonas* [31]. The experiments yielded serendipitous results in that no antibacterial properties were observed, but rather, the rats injected with vinblastine died at a similar rate to those injected with cortisone, a known immunosuppressive agent [31]. The similarity of vinblastine’s immunosuppressive effects to those of corticosteroids led to the discovery of vinca alkaloid’s ability to induce lymphocytopenia and was thus the origin of its anticancer potential [31].

The heart of vinblastine’s anticancer properties lies within its unique mechanism of action: vinblastine as well as the rest of the vinca alkaloid family halts cell division and growth by specifically targeting mitotic spindle formation during the mitosis phase of the cell cycle [5,31,32]. During the metaphase of the active dividing stages of the cell cycle, microtubules begin to elongate from the centrosome and connect to sister chromatids at the metaphase plate; it is at this step of spindle elongation in which vinblastine exhibits its inhibitory properties [32]. Vinblastine has extremely high affinity for the beta subunit of the tubulin dimers that compose the finalized microtubule; thus, it acts as a physical blockade for these tubulin dimers to connect to each other and form organized microtubules and rather promotes the formation of unorganized dimer aggregates [5,32]. The buildup of these aggregates, known as paracrystalline tubulin dimer aggregates, ultimately leads to the destruction of already-formed microtubules and the inhibition of future microtubule elongation [5]. Interestingly, while vinblastine and other vinca alkaloids are extremely potent killers of rapidly dividing cells, some resistant cancer cells have developed vinca alkaloid resistance in the form of upregulating a gene responsible for creating P-glycoprotein, a well-known cell membrane transporter [32]. Upregulation of this gene leads to increased efflux of the vinca alkaloid concentration out of cancer cells, thus limiting efficacy [32].

In the clinical setting, vinblastine and its cousins vincristine and vinorelbine are used in the treatment of many cancers, including acute leukemia, lymphoma, Wilms tumors, neuroblastoma, testicular carcinoma, Kaposi sarcoma, non-small-cell lung cancer, and breast cancer [32]. While vinblastine, vincristine, and vinorelbine all operate within the same mechanism of action, patients may be able to tolerate one more than the others based on the drugs’ side effect profiles. The vinca alkaloids are often associated with immunosuppression, opportunistic infections, severe gastrointestinal upset, constipation, neurotoxicity, alopecia, and syndrome of inappropriate antidiuretic hormone secretion (SIADH) [32,33].

In the realm of neuro-oncology, vinblastine possesses the ability to cross the blood–brain barrier, further increasing its potential as a neuro-oncology therapeutic agent. Additionally, vinblastine’s cousin, vincristine, has been a standard-of-care treatment for many cancers affecting both the central and peripheral nervous systems [5]. Specifically, it is used in the combination treatment of procarbazine, lomustine, and vincristine in order to minimize resistance development in cancers [5]. Some examples include IDH (isocitrate dehydrogenase)-mutant oligodendrogliomas and IDH-mutant astrocytomas [5]. However, as stated previously, the vinca alkaloids must be dosed properly due to their potential neurotoxic side effects [5]. Vinca alkaloids are known to induce peripheral neuropathies and inhibit neurite outgrowth [34,35].

Vinblastine’s journey to becoming a standard chemotherapy in the oncologist’s medical toolbox has been a uniquely winding one, from starting as a diabetes medication and failing as an antibiotic to fully realizing its potential as an anticancer agent. 

### 2.3. Procarbazine 

The initial synthesis of the prodrug procarbazine prompted the idea of its original function being that of a monoamine oxidase inhibitor (MAOI); after all, procarbazine’s structure largely resembles other members of the MAOI family [36]. The purpose of MAOIs is to prevent reuptake of neurotransmitters after they are released into the synaptic cleft between neighboring neurons [37]. This family of drugs is known to be the first class of antidepressants, though today they are seldom used as the standard of care due to dietary restrictions and a lengthy side effect profile [37]. Since procarbazine does share a similar chemical structure to other MAOIs, it does have limited MAOI activity, though it is weaker than many of its family members. Its weak MAOI activity is important clinically, however, because its MAOI activity must be considered in patients that may be taking complex medical regimens; procarbazine has been shown to act synergistically with other MAOIs, leading to neurotoxic levels of neurotransmitter release [36]. In high doses, along with neurotoxicity, it has been shown that procarbazine can also lead to myelosuppression [36,38]. In trial experiments conducted by Bollag and Grunberg testing various hydrazines on cancer lines, it was serendipitously discovered that a potent hydrazine derivative, 1-ethyl-2-benzyl-hydrazine, had potent antitumor activity [39]. This discovery inspired Mathe et al. to test related hydrazine derivatives for enhanced antitumor activity, in which benzyl-hydrazine chlorhydrate (also known as Natulan or procarbazine) was discovered specifically in the context of treating patients with Hodgkin lymphoma [39].

Procarbazine was also shown to cross the BBB and alter the CNS environment [38]. Merging these discoveries and observing its alkylating myelosuppressive effects led clinicians to explore it as a potential CNS tumor chemotherapy drug [36,38]. While it is known that procarbazine contains a potent anticancer effect via DNA alkylation, its exact mechanism of action is still being investigated [7]. Procarbazine has been shown to be a cross-linker between guanine nitrogenous bases, and cross-linkers have been shown to inhibit DNA strand separation [40]. In addition to alkylating DNA, procarbazine has been shown to have other negative effects toward rapidly dividing cells, such as inhibiting the transmethylation of methionine onto loading transfer ribonucleic acid (tRNA), prolonging interphase of the cell cycle, and inducing fatal chromosome breakages [40]. Since procarbazine is also given as a prodrug and must undergo first-pass metabolism, a critical downstream metabolite is produced: hydrogen peroxide [40,41]. Hydrogen peroxide is a free oxygen radical, a family of highly reactive oxygen species that can wreak havoc on DNA and induce mutagenesis [40,41]. Having this arsenal of cell-killing mechanisms allows procarbazine to be a potent chemotherapeutic agent against malignancies such as gliomas, Hodgkin lymphoma, and non-Hodgkin lymphoma [40].

Clinically, procarbazine is used in combination therapy alongside other chemotherapeutic agents to minimize resistance development. In the realm of neuro-oncology, procarbazine is utilized with vinblastine and lomustine in PCV to treat recurrent high-grade glioma [5,42]. In a study comparing an alternative treatment option for recurrent high-grade glioma, i.e., temozolomide (TMZ), PCV was not found to be better or worse than TMZ, meaning that either option could be considered by clinicians [42]. In another study comparing PCV to TMZ, the toxicity profile, overall survival (OS), and time-to-progression were determined in both treatment arms [43]. However, it was observed that TMZ did elicit a slight increase in quality of life and patient performance compared to PCV [43]. A separate phase II trial tested the efficacy of procarbazine and Tamoxifen as a second-line post-surgery and -radiation regimen in patients with glioblastoma or anaplastic astrocytoma, which was preceded by initial treatment with a nitrosourea [44]. Additionally, a separate group was tested with pre-treatment of procarbazine ahead of a nitrosourea versus a nitrosourea alone [42]. In all the groups containing procarbazine, it was found that there was an increased response rate in quality of life and patient performance in both glioblastoma and anaplastic astrocytoma patients; however, there were no changes in median survival time or time to progression [44].

As can be seen, procarbazine has the potential to be a useful chemotherapeutic agent against specific CNS tumors. While its benefits for patients have been observed in numerous trials, so have its downsides in the form of its side effects. Procarbazine has been associated with increased secondary white blood cell (WBC) cancer development, bone marrow suppression, central nervous system and peripheral nervous system toxicity, nausea and vomiting, and hemorrhage [36,40]. 

### 2.4. Lomustine 

Lomustine, commonly known as CCNU, is a chemotherapeutic agent used in the treatment of various brain tumors. The development of lomustine as a therapeutic option was significantly influenced by the insights gained from the observation of mustard gas’s effect on WBC counts. Blister agents were widely used in chemical warfare due to their cytotoxic alkylating effects [45]. One of these blister-causing agents, sulfur gas, commonly known as mustard gas, was first used in World War I in 1917. It became one of the deadliest chemical weapons deployed due to its ability to spread through droplets, causing severe lung damage through inhalation and painful blisters upon skin contact through absorption. In 1919, a study by Krumbhaar et al. examined bone marrow tissues from autopsies on mustard gas victims and concluded that mustard gas plays a role in suppressing the development and maturation of hematopoietic cells and bone marrow tissues, leading to leukopenia. Nitrogen mustards were developed years later as a potential chemical weapon [46]. Although they were never used as weapons, nitrogen mustards surprisingly elicited similar results in Hodgkin lymphoma, lymphosarcoma, and leukemia [47].

This discovery led to the use of nitrogen mustards as one of the first successful chemotherapy agents. This pushed towards the exploration of other alkylating agents, such as nitrosoureas. Lomustine, which belongs to the class of nitrosourea agents, is rapidly absorbed in the gastrointestinal tract and activated in the liver. The active metabolites can then cross the BBB and exert their effects by cross-linking the purine bases within the DNA of rapidly dividing cells, preventing synthesis of DNA, RNA, and protein and, ultimately, promoting cell death [48]. It was approved as a chemotherapy treatment in 1977 and is currently used as second-line treatment for patients with Hodgkin lymphoma and non-Hodgkin lymphoma as well as for childhood gliomas [49]. 

### 2.5. Etoposide 

Etoposide, derived mainly from the plant *Podophyllum peltatum*, is a widely used treatment for various cancers. However, before the discovery of its effectiveness as a chemotherapy drug, extracts of *Podophyllum* were used by various cultures, including Native Americans in the United States, and various groups across the Himalayan region and Western China as a laxative and a remedy for tuberculosis, psoriasis, syphilis, and warts [50,51]. This led to studies that investigated the antiviral effect of these extracts. It was found that the purified podophyllotoxin component of the extract inhibits the replication of various viruses, including measles, herpes simplex type I, cytomegalovirus, Sindhbis, and human papilloma viruses [52]. These early studies then led to the discovery of podophyllotoxin’s antimitotic effect on cells and an unexpected shift in its use as a therapeutic agent in cancer therapy.

Between the years 1947 and 1949, the cytotoxic effects of the podophyllotoxin were accidentally discovered, and it was later confirmed that podophyllin was lethal to mouse sarcomata and to lung tumors in in vitro experiments [53]. In 1949, a study by Sullivan et al. was the first to use podophyllin resin, a less purified and more stable form of podophyllotoxin, to treat skin carcinomas. Similar cell-damaging effects were observed by earlier researchers, and results showed that locally applied podophyllin was a treatment for human basal-cell carcinoma [54]. In 1966, etoposide was synthesized from *Podophyllum*. It showed effectiveness in inhibiting topoisomerase II, leading to DNA strand breakage and arresting of cells in the late S/G2 phases of the cell cycle [55]. These discoveries and developments underscore the significant contributions of *Podophyllum* and its derivatives in advancing cancer treatment, demonstrating their lasting impact on the field of oncology.

Clinical trials of etoposide’s use began in 1971 as it demonstrated its effectiveness in treating cancers including small cell lung cancer, leukemias, lymphomas, ovarian cancer, and Kaposi’s sarcoma [56]. In 1983, etoposide was approved by the FDA to treat testicular cancer [57]. To date, multiple studies have shown etoposide as a potential treatment for patients with malignant brain tumors including glioblastoma multiforme (GBM) due to its ability to enhance the permeability of the BBB and exert its effects [58,59,60]. In a series of studies by Chamberlain, oral etoposide was shown to have a response rate of 50% to 63% in children and young adults who have relapsed or recurrent medulloblastoma and glioma [61]. Currently, etoposide has only gained U.S. Food and Drug Administration (FDA) approval for refractory testicular tumors and small cell lung cancer. Although the use of etoposide has shown much promise in combination therapies against brain cancers, more research is needed to push approval for etoposide as a primary treatment.

### 2.6. Avastin 

Avastin, or bevacizumab, is a monoclonal antibody directed against vascular endothelial growth factor (VEGF). It is used in the treatment of various cancers, including colorectal, hepatocellular, cervical, renal cell cancers, and glioblastoma [62]. Napoleone Ferrera and his team at Genentech first set out to explore VEGF after it was isolated in his lab in 1989 [63]. Over the next few years, the properties of VEGF in molecular biology and biochemistry were researched, including its receptors, regulations, and genetic isoforms [64,65,66,67]. When it came time to assess VEGF’s function in mice, its crucial role in vasculogenesis and development was discovered. VEGF has well-described roles in bone formation, corpus luteum maintenance, ovarian maintenance, pancreatic development, and retinopathy [68,69,70,71,72,73].

Discovering that VEGF drives many developmental processes, inducing endothelial cells to produce survival factors and promote growth, led to hypotheses about creating agonists to use protectively in diseased tissue and antagonists to slow undesirable tissue growth. It was believed that by inhibiting VEGF in patients with tumors, slowed angiogenesis would delay tumor growth. In vivo trials showed that anti-VEGF antibodies had an inhibitory effect on glioblastoma multiforme-, leiomyosarcoma-, and rhabdomyosarcoma-bearing mice. This was surprising at the time, considering it did not have a similar effect in tumor cell lines in vitro [74,75]. As in vivo tumor trials for ovarian, colon, breast, and prostate cancers became successful, further studies confirmed that tumor blood vessel density was significantly decreased in mice treated with anti-VEGF antibodies [74,76,77,78,79,80]. This supported the hypothesis that slowed angiogenesis delays tumor growth. Once human tumor cell lines implanted in mice were shown to be treatable with anti-VEGF antibodies, bevacizumab was developed [81,82]. After bevacizumab was developed and found effective in vivo, it advanced to clinical trials [81]. Phase 1 trials showed the antibody to be non-toxic, and phase 2 trials showed effective treatment in metastatic colorectal cancer, non-small-cell lung cancer, and renal cell cancer [83,84,85,86,87]. The drug was approved by the FDA in 2004 for metastatic colorectal cancer, and its approval for other cancers followed. Avastin’s arrival to neuro-oncology was planned rather than serendipitous after it was discovered that glioblastoma is a tumor that secretes high amounts of VEGF. Since bevacizumab had success in other malignancies with high VEGF profiles, including metastatic brain cancers, trials in glioblastoma followed soon after its approval [88].

Once Avastin arrived on the scene, glioblastoma treatment would never be the same. VEGF is a glioblastoma-associated biomarker, and its level has a direct correlation with patient survival [89]. This led to the hypothesis that bevacizumab could slow tumor growth and improve survival in glioblastoma patients by inhibiting VEGF. When tried in recurrent glioblastoma patients, bevacizumab was safe and had strong blood–brain penetration; when tried in progressive glioblastoma patients, it improved progression-free survival [90,91]. In newly diagnosed patients, adding bevacizumab to radiotherapy and temozolomide did not improve survival but had a significant effect in progression-free survival and quality of life [92,93]. As GBM treatment advances, Avastin’s efficacy has remained constant. It has been used in trials alongside epidermal growth factor receptor (EGFR) inhibitors and other immune checkpoint inhibitors, with some success in treating glioblastoma [93,94,95,96,97]. Additionally, epidemiologic data have shown that since the approval of bevacizumab, overall survival and median survival have all significantly improved in glioblastoma patients [98,99]. By inhibiting VEGF, Avastin has had a large impact in limiting tumor growth and has allowed many patients to continue enjoying their lives. This highlights the drug’s practicality in treatment and foreshadows a new era in glioblastoma treatment.

Avastin’s success demonstrates the impact of the molecular era of targeted antibody treatments. Continued research of unique molecular targets in various malignancies has led to the development of different antibodies targeting markers like CD20, EGFR, and more [93]. Alongside the identification of these markers is the development of nanobodies to improve delivery of these antibodies and further revolutionize the landscape of neuro-oncology. As our understanding and discoveries of molecular markers and checkpoints for tumors increase, specific treatments like Avastin will become available to target these and effectively treat previously difficult-to-treat malignancies [93].

### 2.7. Temozolomide 

Temozolomide was first invented in the 1970s as researchers aimed to synthesize compounds with antitumor effects. Temozolomide did not stand out at first, but as it was continuously developed and tried, it was later found to have a unique ability to safely penetrate the blood–brain barrier [100]. Before temozolomide became approved for treatment of glioblastoma in 2005, the disease had an abysmal prognosis. Currently, treatment options for glioblastoma include radiotherapy, surgical resection, and temozolomide chemotherapy [101]. There was some past success in improving survival using other alkylating agents. However, due to the high doses needed to penetrate the BBB, these regimens were too toxic to be beneficial. They left a high rate of recurrence, and most patients were unable to tolerate repeat therapy due to initial toxicities [102].

Temozolomide revolutionized the treatment of glioblastoma because, as an alkylating agent, it can rapidly penetrate the cerebrospinal fluid, does not require hepatic metabolism, and has 100% bioavailability with linear pharmacokinetics [103]. This has allowed for non-toxic treatment of glioblastoma [104]. Before temozolomide was approved, patients treated with surgery and radiation had a median survival of 12 months; since the temozolomide era, median survival has improved to 19 months. Notably, younger patients aged 20–29 had an overall survival up to 32 months [101,105,106]. Temozolomide has improved the outcome for GBM patients significantly, but it is not perfect, and research into advancing treatment is ongoing. One issue is resistance, which develops in nearly half of patients. This occurs when mutations in the tumor cause it to evade the suppressive effects of the drug. These mutations include those that encode histone demethylase, epidermal growth factor receptors, and DNA repair mechanisms like mismatch repair pathway or base excision repair [106,107,108,109,110,111]. Efforts in improving treatment aim to find a regimen to synergistically suppress GBM tumors and the various cellular pathways that drive their growth and resistance. At the center of these regimens is temozolomide. 

## 3. Out of the Box 

In neuro-oncology, several unconventional and unlikely treatments have shown promising antitumor effects. A few of these include the use of Boswellia, thalidomide, high-dose tamoxifen, Celebrex, Viagra, Accutane (cis-retinoic acid), Tarceva, and Gleevec combined with hydroxyurea. Although these drugs provide insight to potential mechanisms of treating brain malignancies, they lack the clinical significance and results to be of use in practice. Other drugs in this category that are not mentioned in this review include proton pump inhibitors, disulfram, rapamycin, metformin, lonidamine, chloroquine, and chloropromazine (Table 1).

### 3.1. Accutane 

Since its FDA approval in 1982, Accutane, also known as isotretinoin, is a synthetic derivative of vitamin A that remains the most effective treatment for severe acne due its ability to reduce sebum production, promote anti-inflammatory effects, and also lower levels of *Propionibacterium acnes* [112]. Beyond dermatological applications, isotretinoin has been repurposed for neuro-oncology, particularly as an off-label treatment for high-risk neuroblastoma (NB) [113]. Surprisingly, in vitro studies have demonstrated isotretinoin’s chemotherapeutic effects, sustained growth arrest, and ability to induce differentiation in neuroblastoma cell lines [113]. Additionally, due to its lipophilic properties, Accutane has the ability to cross the BBB [114]. However, its effectiveness is limited to preventing the progression of NB, and it has not been shown effective in treating the tumor itself [115]. As a result, studies demonstrating its effectiveness in treating tumors may be needed before Accutane can become a standard treatment for neuroblastoma. 

### 3.2. Sildenafil 

Sildenafil, most commonly known as brand-name Viagra, is a phosphodiesterase5 inhibitor (PDE5-I) most well known for its use for erectile dysfunction. Sildenafil was originally designed for treatment of angina pectoris. In a 1992 study, some participants reported penile erections as a side effect of the drug [116]. With this discovery, clinical studies began for sildenafil as treatment for erectile dysfunction [116]. In vitro and in vivo studies discovered sildenafil’s surprising ability to enhance the efficacy of other chemotherapy agents to treat various cancers by inducing apoptotic effects, arresting the cell cycle, and increasing production of reactive oxygen species [117]. As a selective PDE5-1, sildenafil can exert its apoptotic effects in GBM by preventing the degradation of cyclic guanosine monophosphate (cGMP) and activating protein kinase G (PKG), which in turn downregulates anti-apoptotic proteins such as B-cell lymphoma 2 [118]. Furthermore, sildenafil is a moderately lipophilic drug, and studies have shown its ability to cross the BBB and interact with PDE5 expressed in brain cells [119,120]. 

Additionally, an ongoing phase 2 clinical study that combined sildenafil with sorafenib tosylate and valproic acid for treatment of patients with recurrent high-grade glioma has shown promise for sildenafil as an adjunctive therapy [121]. Therefore, further studies are needed to confirm sildenafil as a possible treatment in gliomas. 

### 3.3. Thalidomide 

Thalidomide, a racemic derivative of glutamic acid, initially gained popularity in 1956 for its use as a sedative without the associated side effects of dependency or a hangover. Soon, thalidomide was prescribed as an antiemetic for women suffering with nausea and vomiting during the first trimester of pregnancy [122]. However, in 1961, Dr. William McBride and Dr. Widukind Lenz independently discovered that thalidomide used in pregnancy can result in congenital malformations [123,124]. Soon after the teratogenic properties were reported, research focusing on antitumor properties of thalidomide began.

In 1994, a study by D’Amato et al. discovered that thalidomide can inhibit the formation of new blood vessels, which spurred research into its use as an anti-angiogenic agent [125]. It was found that thalidomide may have potential as a treatment for hematologic malignancies including plasma cell myeloma, myelodysplastic syndromes, myelofibrosis, and macroglobulinemia [126]. In neuro-oncology, studies have also been undertaken to utilize thalidomide’s anti-angiogenic effect in patients with high-grade gliomas [127]. Additionally, with thalidomide’s small structure and its ability to diffuse passively through tight junctions, it is able to readily cross the BBB, making thalidomide a potential treatment in brain cancer [128]. However, at present, thalidomide has not been shown useful as a monotherapy for high-grade gliomas. 

### 3.4. High-Dose Tamoxifen 

Tamoxifen is a selective estrogen receptor modulator (SERM) indicated for breast cancer treatment [129]. It can block 17-beta-estradiol (E2) at the receptor site but also cause carcinogenesis [130]. A study by Lien et al. was the first to report that tamoxifen accumulates in normal brain tissue at surprising levels of up to 46-fold higher than those found in serum. This high tissue concentration is attributed to tamoxifen’s low molecular weight and lipophilicity, which enables it to efficiently cross the BBB [131].

In addition, studies have demonstrated that activation of protein kinase C (PKC) can play a role in the growth of high-grade gliomas [132]. It was also shown that the inhibition of PKC can increase the effect of radiation therapy [133]. In a phase 2 trial by Robins et al., a high dose of tamoxifen was administered during and after radiation. It compared the overall survival of 1457 patients from the Radiation Therapy Oncology Group (RTOG) database of past GBM studies. The median survival time was 11.3 months, which did not significantly differ from previous studies (*p* = 0.94) [133]. Thus, further studies of tamoxifen are needed to show improved survival outcomes in patients with neurological malignancies. 

### 3.5. Celebrex 

Celebrex, also known as celecoxib, is a nonsteroidal anti-inflammatory drug (NSAID) that was designed to selectively inhibit the cyclooxygenase-2 (COX-2) isomer of the cyclooxygenase (COX) enzyme [134]. In particular, COX-2 is activated by cytokines and mitogens to form prostaglandins, promoting inflammation. Additionally, overexpression of COX-2 has been found to lead to tumorgenesis [135]. In 1998, celecoxib was the only COX-2 inhibitor approved by the FDA for rheumatoid arthritis and osteoarthritis. Clinical trials began soon thereafter, examining celecoxib as a potential chemo-preventative agent [136]. A study by Reddy et al. showed with statistical significance that celecoxib can both inhibit the incidence of adenocarcinoma of the colon and decrease its multiplicity in a dose-dependent manner [137]. However, investigations into the side effects of COX-2 inhibitors began when the drugs were linked to an increased risk of cardiovascular diseases such as myocardial infarction, stroke, and heart failure [138].

In the context of neuro-oncology, a study by Kang et al. showed that celecoxib was surprisingly able to decrease cell viability of malignant GBM cells [139]. Additionally, a study by Novakova et al. showed that due to celecoxib’s ability to passively diffuse across the BBB, it is able to accumulate in the brain [140]. More recently, a study by Yin et al. demonstrated that when administered with TMZ, the combination therapy can inhibit cell proliferation of TMZ-resistant GBM cell lines [141]. Despite these advancements, further studies evaluating the efficacy and safety of celecoxib should be conducted before it may be used as treatment for cancer. 

### 3.6. Gleevec and Hydroxyurea 

Gleevec, the brand name of imatinib, is a tyrosine kinase inhibitor that was initially FDA-approved in 2001 to treat Philadelphia chromosome-positive chronic myelogenous leukemia (Ph+ CML) [142]. Tyrosine kinases are involved in the signaling cascade and, when activated by ATP, catalyze the protein tyrosine phosphorylation on its substrates [143]. Gleevec acts by binding near the ATP binding site, ultimately inhibiting the enzymatic activity of tyrosine kinase [143]. Additionally, while imatinib does not easily cross the BBB, in areas where the BBB is disrupted, it can accumulate at intratumoral levels comparable to or even exceeding those found in plasma [144]. In contrast, hydroxyurea is an antimetabolite, diffusing into cells, where it blocks the active site of ribonucleotide reductase, inhibiting its activity [145]. Currently, it is approved for treatment of sickle cell anemia, CML, and certain types of head and neck cancers [146].

In neuro-oncology, a study by Raymond et al. showed what they termed “pseudo-improvements”, as patients with glioma who received imatinib showed a decrease in contrast enhancement on magnetic resonance imaging (MRI) [147]. While it may suggest an increase in efficacy for the drug, further testing showed that there was no change in the tumor size. Instead, this reduction in contrast enhancement was attributed to imatinib’s ability to normalize tumor vasculature, reducing blood vessel leakiness [147]. Though initially viewed as a misleading treatment response, this finding highlighted imatinib’s role in stabilizing the tumor microenvironment. This insight has since increased research into vascular normalization as a potential therapeutic approach for gliomas.

Additionally, a clinical study by Dresemman showed that in 57% of treatment-refractory GBM patients, an imatinib and hydroxyurea combination treatment produced a partial or complete response or resulted in stable disease lasting at least three months [148]. However, use of hydroxyurea has been limited due to its toxic side effects and low therapeutic index. Hydroxyurea has been reported to cause anemia, leukemia, skin cancers, and gastrointestinal conditions, to name a few [149]. At present, the combination treatment lacks clinical evidence demonstrating its effectiveness in treating cancers, including GBM. Consequently, it has not become a standard treatment in neuro-oncology. 

### 3.7. Tarceva 

Tarceva, a small-molecule inhibitor also known as erlotinib, was first developed by OSI Pharmaceuticals and Genentech in 2004 for treatment of metastatic non-small-cell lung cancer (NSCLC), pancreatic cancer, colorectal cancer, and head and neck cancer [150,151,152]. It was serendipitously accepted into the exclusive FDA’s Pilot 1 Program, an extension of Fast Track, allowing for drug developers to send segments of their drug acceptance applications as completed to the FDA with the goal of speeding up the approval process [150,151]. This program is specifically reserved for promising novel pharmaceuticals that fulfill an unmet clinical problem [150,151]. Tarceva first exhibited its therapeutic potential in its phase III trial for patients with NSCLC and elicited improvement in the metrics of overall survival, progression-free survival, and response rate [153]. Tarceva is a reversible, highly selective small-molecule inhibitor for cells expressing epidermal growth factor (HER1) tyrosine kinase receptors and possesses the capability of crossing the BBB [150,151,152]. Specifically, Tarceva fits in a notch within the receptor that physically blocks ATP from entering, therefore blocking the signal transduction pathway at the receptor level, resulting in mitigated cellular growth and proliferation [151,153,154]. Despite the efficacy shown in Tarceva’s phase III trial, resistance in a variety of cancers has been elicited in multiple instances, especially in NSCLC [154]. In patients with NSCLC being treated with Tarceva, resistance is typically observed after eight months. Genetic studies have found that the most common epidermal growth factor receptor (EGFR) mutation is T790M, a threonine-to-methionine substitution at position 790, of the EGFR gene, which leads to looser binding of Tarceva within the ATP notch [154].

In addition to the malignancies listed above, Tarceva has been tested in other EGFR+ malignancies, specifically within the realm of neurological malignancies. However, little clinical efficacy has been observed. In a phase II trial testing, Tarceva in combination with carboplatin was tested for patients with less than two relapsed glioblastoma multiforme, and no correlation was observed in the Tarceva + carboplatin arm for progression-free survival or overall survival compared to the placebo [155]. In another phase I/II trial testing Tarceva and temsirolimus in patients with glioblastoma or anaplastic glioma, little therapeutic efficacy was observed in comparison to the non-treatment arm; it was hypothesized that the lack of clinical significance could be due to too-low drug concentrations due to the toxicities observed or because the signaling pathways targeted were too repetitive, driving mutagenesis and resulting in drug resistance [156]. 

### 3.8. Boswellia 

Boswellia, most known as frankincense, is an herbal extract of the resin made from the bark of the *Boswellia serrata* tree. Typically, the gum resins of the Boswellia species have been used as adhesives, incense, and cosmetics [157]. However, the natural compounds present have also been shown to have antimicrobial, antiviral, and anti-inflammatory effects. Additionally, isolated boswellic acids, extracted from Boswellia, have also been shown to have potential as anticancer treatments [158]. In a study by Glaser et al., boswellic acids were shown to induce apoptosis in patients with malignant glioma [159]. Acetyl-11-keto-beta-boswellic acid (AKBA) specifically enhances CD95, a cell-surface protein, to induce apoptosis in human glioma cells. Additionally, due to its high lipophilic property, Boswellia can readily cross the BBB, making it a potential treatment in neuro-oncology.

**Table 1 ijms-26-02955-t001:** Out-of-the-Box Therapies not described.

Drug (Year of Discovery)	General Mechanism of Action	Mechanism in Neuro-Oncology
**Proton Pump Inhibitors** (1980)	Blocks the gastric H^+^/K^+^-ATPase to decrease acid secretion	May disrupt tumor pH regulation—by inhibiting vacuolar-type H^+^-ATPases in cancer cells—which can alter drug uptake/resistance and potentially sensitize tumors (e.g., gliomas) [160,161]
**Disulfiram** (1881)	Inhibits aldehyde dehydrogenase and modulates cellular redox balance; also affects proteasome and NF-κB signaling	Repurposed to target cancer stem cells (including in glioblastoma) via ALDH inhibition and copper complex formation that increases oxidative stress in tumor cells [162,163]
**Rapamycin** (1975)	Binds FKBP12 to inhibit mTOR signaling, thereby reducing cell growth and inducing autophagy	Inhibits mTOR—a pathway often hyperactive in gliomas—to suppress tumor cell proliferation, reduce angiogenesis, and modulate autophagy in brain tumors [164]
**Metformin** (1922)	Activates AMP-activated protein kinase (AMPK) to lower hepatic gluconeogenesis and modulate cellular energy metabolism	In neuro-oncology, AMPK activation leads to indirect mTOR inhibition and decreased tumor cell proliferation, with preclinical studies suggesting antiglioma effects [165]
**Lonidamine** (1970s)	Inhibits aerobic glycolysis and disrupts mitochondrial energy metabolism in cancer cells	Alters the energy metabolism of tumor cells—including glioma cells—potentially enhancing sensitivity to chemotherapy by targeting the glycolytic pathway [166]
**Chloroquine** (1934)	Raises lysosomal pH and blocks autophagy, with additional immunomodulatory actions	By inhibiting autophagy, it can compromise tumor cell survival and may be used to enhance the effects of chemo- and radiotherapy in brain tumors such as glioblastoma [167]
**Chlorpromazine** (1950)	Antagonizes dopamine receptors (with additional effects on multiple neurotransmitter systems)	Has been reported to induce apoptosis and interfere with signaling pathways in glioma cells—suggesting potential repurposing as an adjuvant agent in neuro-oncology [168]

## 4. Conclusions

The future of neuro-oncology lies in continuing to overcome many obstacles, such as the blood–brain barrier, through newer, more innovative drug delivery mechanisms, perhaps using combination therapies as a way to treat a tumor from multiple angles. 

It is important to recognize that the history of neuro-oncology drug development is full of serendipitous discoveries. Given the challenges of developing new drugs, repurposing existing drugs may improve survival rates and the quality of life of patients with CNS tumors. Research studying unexpected side effects of medications initially intended for other uses has led to advances in neuro-oncology treatments, and navigating through the history and development of these medications highlights the serendipitous nature of medical science. 

## Figures and Tables

**Figure 1 ijms-26-02955-f001:**
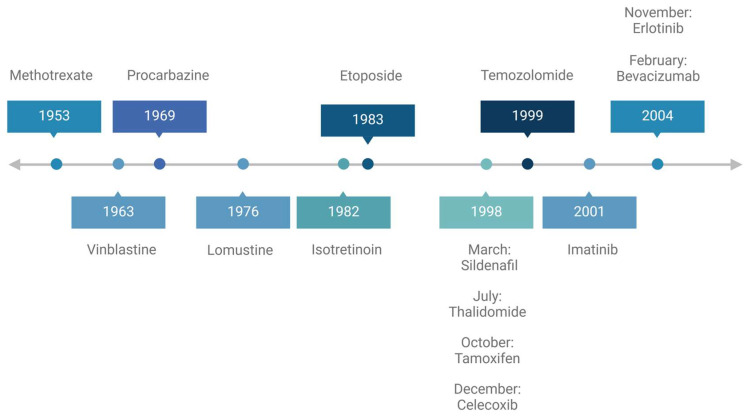
Timeline of FDA approval.

## Data Availability

No new data were created or analyzed in this study. Data sharing is not applicable to this article.

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
