# Peer review of "Serendipity in Neuro-Oncology: The Evolution of Chemotherapeutic Agents"

_ijms, 2025, doi:10.3390/ijms26072955_

Round 1
Reviewer 1 Report
Comments and Suggestions for Authors
My comments are attached

Author Response
We would like to thank the reviewers for their constructive feedback. Implementing the recommended edits had definitely made the manuscript better. Please find below our response to comments from reviewers:
Reviewer Comments:
Minor:
- The affiliation of each author should be elaborated in detail and should just not include the name of the university
We have edited the affiliation of each author on the manuscript.
- Please define the full forms of acronyms at the first instance. For example, B-HCG (Page 2),
SIADH (Page 3), PCV, and CCNU (Page 4)
We have edited the manuscript with the full forms of the acronyms at first instance.
- Please correct Hodkin Lymphoma to ‘Hodgkin’ on page 4
We thank the reviewer for highlighting this important point. We have corrected Hodkin to Hodgkin on the manuscript.
Major:
- While the resistance offered by the natural immune system to the therapeutics plays a major role in insufficient drug delivery to the brain, the major barrier in the neurooncology is the resistance offered by the blood-brain barrier which reduces a significant amount of the drug reaching the target tumor. I think this needs to be elaborated in detail in the introduction and correlated with the physicochemical properties of the drugs that were repurposed in particular addressing this concern.
We thank the reviewer for their comment. We have edited the manuscript to discuss blood-brain barrier.
- Could the authors elaborate on what does response rate scale indicates? For instance, line 63 says the response rates up to 70.
We have added what response rate indicates.
- It is vital to mention the discovered role of folate for instance its role in cell growth and how were they targeted to treat cancer. Also, the serendipitous discovery story of Methotrexate to treat it.
We have edited the manuscript to discuss folate and the serendipitous discovery of methotrexate.
- Could the authors elaborate on how Procarbazine was a serendipitous discovery? Since the description only makes a mention of it being a structural mimic of MAOI.
We thank the reviewer for bringing this to our attention. We have edited the manuscript to elaborate how procarbazine was a serendipitous discovery.
- Since the title says serendipitous discovery of the drugs in neuro-oncology, it will be useful to incorporate the serendipitous discovery story of each drug in neuro-oncology instead of elaborating on the anti-cancer effect of each drug in different cancers which appears as a diversion from the major goal.
We have edited the manuscript to address this comment.
- It will be interesting to evaluate the serendipitous identification of different anti-cancer targets and the discovery of other novel antibodies/ checkpoint inhibitors in the review except Avastin.
After reviewing novel antibodies, we found that bevacizumab emerged as the leading new treatment primarily due to its comparatively milder side effect profile. However, our team did not find other antibodies to be clinically significant to add on to our manuscript.
https://pmc.ncbi.nlm.nih.gov/articles/PMC8524703/
- Although the discovery of the role of VEGF might be serendipitous, the development of anti VEGF antibodies was a planned attempt. Could, the authors elaborate on the serendipity of Avastin discovery related to glioblastoma?
We have edited the manuscript to address this comment.
- The description of Temozolomide appears to be incomplete.
We have edited the manuscript to address this comment.
- It will be interesting to study how Accutane was repurposed for neuro-oncology applications.
We have edited the manuscript to address this comment.
- It will be interesting to understand the mechanism by which Sildenafil induces an apoptotic effect.
We have edited the manuscript to address this comment.
- Majority of the out of the out-of-the-box drugs repurposed for cancer are rather proposed as adjunctive therapy. It will be vital to illustrate their description in the form of a table and only those which have direct anti-neurooncological potential should be elaborated in the description.
We appreciate the suggestion to further categorize repurposed drugs based on their role in cancer therapy. However, the essence of this portion of the manuscript is to highlight how these medications defy conventional classification, which is why we refer to them as "out-of-the-box" therapies. Placing them into predefined categories would not fully capture the evolving and unexpected ways they have been applied in neuro-oncology. Instead, we believe that focusing on key examples best conveys their impact.
- A lot of drugs including proton pump inhibitors, Disulfuram, Rapamycin, Metformin, Lonidamine, Chloroquine and other anti-malarial derivatives, Chlorpromazine, and other dopamine receptor inhibitors were serendipitously discovered for the treatment of glioblastoma which is missing in the review. It will be better if the authors group all the drugs into structurally similar categories and then draw conclusions on their application in neuro-oncology which will be useful for wider readership of the present manuscript. Or an elaborate table could be mentioned enlisting the drugs, year of serendipitous discovery and their mechanism of action.
We carefully considered these drugs, along with several others, in our evaluation. After thorough review, our team concluded that they were not as impactful or clinically significant as the medications discussed in our manuscript. Below are the key articles that informed our decision:
Chlorpromazine https://pubmed.ncbi.nlm.nih.gov/38162492/
Metformin https://pubmed.ncbi.nlm.nih.gov/36907105/
Disulfuram https://jamanetwork.com/journals/jamanetworkopen/fullarticle/2802966
Rapamycin https://www.dovepress.com/rapamycin-inhibits-glioma-cells-growth-and-promotes-autophagy-by-mir-2-peer-reviewed-fulltext-article-CMAR
Lonidamide https://translational-medicine.biomedcentral.com/articles/10.1186/s12967-023-04332-y
Chloroquine https://www.frontiersin.org/journals/oncology/articles/10.3389/fonc.2018.00335/full
Additionally, while we appreciate the suggestion to include a table outlining the mechanisms of action, we believe that the primary focus of our manuscript is to illustrate the serendipitous discovery and the paradigm shift in how these drugs have been utilized in neuro-oncology. Given this narrative approach, we feel that adding a table with mechanisms of action would not substantially enhance the manuscript’s overall contribution.
- Section 3.6. The last line appears to be incomplete.
We have edited the manuscript to address this comment.
Reviewer 2 Report
Comments and Suggestions for Authors
This paper by Nadora et al., Serendipity in Neuro-Oncology: The Evolution of Chemothera-2 peutic Agents, provides an insightful and timely review of the challenges and opportunities in neuro-oncology drug discovery, particularly in the context of gliomas. The authors argue that, despite significant advances in molecular biology and genomics, targeted therapies have had limited success due to the multigenic and complex nature of brain tumors. The review highlights the role of serendipity in therapeutic breakthroughs, including the unexpected efficacy of repurposed drugs and beneficial off-target effects. It proposes that combining rational drug design with the potential for serendipitous discoveries could improve future drug development in neuro-oncology. Overall, this paper this paper is thematically suitable for publication in IJMS. However, some major concerns should be addressed prior to publication.
Major comments:
1. This review gives a concise overview of the progress of drugs used in neurological tumors. However, as current treatments have had mild success, I suggest that in order to make this review of greater value, the authors should include a section for more innovative drugs that have been developed, for example, Gliadel and molecular drugs.
2. The authors should add a future perspective section to discuss potential breakthroughs and directions further.
Author Response
Reviewer Comments:
Major comments:
- This review gives a concise overview of the progress of drugs used in neurological tumors. However, as current treatments have had mild success, I suggest that in order to make this review of greater value, the authors should include a section for more innovative drugs that have been developed, for example, Gliadel and molecular drugs.
We thank the reviewer for highlighting this important point. We carefully considered these drugs, along with several others such as Metformin, Lonidamine, Chloroquine, in our evaluation. After thorough review, our team concluded that they were not as impactful or clinically significant as the medications discussed in our manuscript.
- The authors should add a future perspective section to discuss potential breakthroughs and directions further.
We have edited the manuscript to address this comment.
Round 2
Reviewer 1 Report
Comments and Suggestions for Authors
Attached

Author Response
We sincerely appreciate the reviewer’s thoughtful suggestions and detailed feedback, which have greatly contributed to strengthening the depth, clarity, and readability of the manuscript. We have carefully addressed each comment and made the necessary revisions to enhance the manuscript. Please let us know if there are any additional refinements recommended:
- Although my previous comment was partially addressed and the blood brain barrier
(BBB) was described in a line or 2. I think the idea of incorporating the BBB was to
correlate the the physicochemical properties of the drugs that were repurposed which
could overcome the challenge with the BBB.
We thank the reviewer for their insightful comment. We appreciate the opportunity to further clarify the connection between the physicochemical properties of the repurposed drugs and their ability to cross the BBB. In response, we have ensured that each drug’s interaction with the BBB is explicitly mentioned in the manuscript. The following sections highlight this aspect:
- Methotrexate: Lines 71-74
- Vinblastine: 142-144
- Procarbazine: 179
- Lomustine: 240-243
- Etoposide: 275-277
- Avastin: 321-323
- Temozolomide: 345-348
- Accutane: 391-392
- Sildenafil: 408-410
- Thalidomide: 432-434
- Tamoxifen: 442-443
- Celebrex: 469-471
- Gleevec and hydroxyurea: 483-485
- Tarceva: 521
- Boswellia: 553-555
Additionally, we have refined the discussion to better emphasize how these drugs' properties contribute to their ability to penetrate the BBB, aiding their effectiveness in neuro-oncology applications. We hope this revision fully addresses the reviewer’s concern, and we appreciate their thoughtful feedback.
- The previous comment 5-‘Since the title says serendipitous discovery of the drugs in
neuro-oncology, it will be useful to incorporate the serendipitous discovery story of
each drug in neuro-oncology instead of elaborating on the anti-cancer effect of each
drug in different cancers which appears as a diversion from the major goal’. The
comment was only addressed for Etoposide. It needs to be addressed for all the drugs
in the interest of the readers and to justify the title of the manuscript.
We greatly appreciate the reviewer’s insight. In response to the reviewer’s suggestion, we have revised the manuscript to highlight the serendipitous discoveries related to each drug within the context of neuro-oncology. The following sections now explicitly emphasize how each drug’s neuro-oncology application was uncovered, ensuring a consistent narrative that aligns with the manuscript’s title:
- Methotrexate: 69-71
- Vinblastine: 107-112
- Procarbazine: 171-174
- Lomustine: 234-238
- Etoposide: 256-258
- Avastin: 312-316
- Temozolomide: 345-348
- Accutane: 389-391
- Sildenafil:402-405
- Thalidomide: 426-428
- Tamoxifen: 440-441
- Celebrex: 468-469
- Gleevec and hydroxyurea: 490-498
- Tarceva: 513-516
- Boswellia: 550-551
By making these revisions, we hope that the manuscript maintains a clear focus on the role of serendipity in neuro-oncology drug discovery, which is central to its theme.
- Previous comment 6-‘ It will be interesting to evaluate the serendipitous identification
of different anti-cancer targets and the discovery of other novel antibodies/ checkpoint
inhibitors in the review except Avastin’ was only partially addressed. The nanobodies
and anti-CD20 antibodies were a serendipity and could be incorporated in the
manuscript to make it a wholesome review.
We thank the reviewer for their suggestion. We agree that the serendipitous discovery of nanobodies and anti-CD20 antibodies plays an important role in the broader landscape of neuro-oncology therapeutics. In response, we have expanded the discussion to better highlight the serendipitous nature of these discoveries and their relevance to neuro-oncology. These additions can now be found in the following lines:
- Nanobodies & Anti-CD20 Antibodies: Lines 335-342
- Previous comment 7- ‘
Although the discovery of the role of VEGF might be serendipitous, the development
of anti VEGF antibodies was a planned attempt. Could, the authors elaborate on the
serendipity of Avastin discovery related to glioblastoma?’ still needs to be addressed in
the manuscript.
We have addressed the serendipitous connection between Avastin and glioblastoma in lines 312-316. However, we understand that additional clarification may be needed. To strengthen this section, we have expanded our discussion to further highlight how Avastin’s use in glioblastoma emerged unexpectedly from its initial development for other malignancies.
If there are specific aspects the reviewer would like us to further elaborate on, we would greatly appreciate their guidance. We value their insightful feedback and are committed to ensuring this aspect is fully addressed.
- The serendipity in the application of Temozolamide to GBM seems to be missing.
We thank the reviewer for their comment. This have been addressed in lines: 426-428, highlighting the unexpected path that led to temozolomide’s use in GBM. If further clarification is needed, we would be happy to expand on specific aspects. We appreciate the reviewer’s valuable feedback.
- I disagree to the response on my previous comment 12- A lot of drugs including proton
pump inhibitors, Disulfuram, Rapamycin, Metformin, Lonidamine, Chloroquine and
other anti-malarial derivatives, Chlorpromazine, and other dopamine receptor
inhibitors were serendipitously discovered for the treatment of glioblastoma which is
missing in the review. It will be better if the authors group all the drugs into structurally
similar categories and then draw conclusions on their application in neuro-oncology
which will be useful for wider readership of the present manuscript. Or an elaborate
table could be mentioned enlisting the drugs, year of serendipitous discovery and their
mechanism of action. Indeed the references in the response by the authors in itself are
evidence that the repurposed drugs are effective and elicit synergistic effect.
We thank the reviewer for their detailed feedback. To address this concern, we have added a comprehensive table summarizing these drugs, including their serendipitous discovery and mechanisms of action, in line 558. This addition enhances clarity and organization, making the information more accessible to a wider readership.
Reviewer 2 Report
Comments and Suggestions for Authors
The authors addressed my concerns, this paper is ready for publication.
Author Response
Thank you, we sincerely appreciate the reviewer’s thoughtful suggestions and detailed feedback.